# Potential of Graphene-Functionalized Titanium Surfaces for Dental Implantology: Systematic Review

Angelo Michele Inchingolo [†], Giuseppina Malcangi [†], Alessio Danilo Inchingolo [†], Antonio Mancini, Giulia Palmieri, Chiara Di Pede, Fabio Piras, Francesco Inchingolo *, Gianna Dipalma *,[‡] and Assunta Patano [‡]

Department of Interdisciplinary Medicine, University of Bari "Aldo Moro", 70124 Bari, Italy; angeloinchingolo@gmail.com (A.M.I.); giuseppinamalcangi@libero.it (G.M.); ad.inchingolo@libero.it (A.D.I.); dr.antonio.mancini@gmail.com (A.M.); giuliapalmieri13@gmail.com (G.P.); c.dipede1@studenti.uniba.it (C.D.P.); dott.fabio.piras@gmail.com (F.P.); assuntapatano@gmail.com (A.P.)
* Correspondence: francesco.inchingolo@uniba.it (F.I.); giannadipalma@tiscali.it (G.D.); Tel.: +39-331-2111-104 (F.I.); +39-339-6989-939 (G.D.)
† These authors contributed equally to this work as first authors.
‡ These authors contributed equally to this work as last authors.

**Abstract:** Titanium is the most frequently employed material in implantology, because of its high degree of biocompatibility. The properties of materials are crucial for osteointegration; therefore, great effort from researchers has been devoted to improving the capabilities of titanium implant surfaces. In this context, graphene oxide represents a promising nanomaterial because of its exceptional physical and chemical qualities. Many authors in recent years have concentrated their research on the use of graphene in biomedical applications such as tissue engineering, antimicrobial materials, and implants. According to recent studies, graphene coatings may considerably increase osteogenic differentiation of bone marrow mesenchymal stem cells in vitro by the regulation of FAK/P38 signaling pathway, and can encourage the osteointegration of dental implants in vivo. However, further studies, especially on human subjects, are necessary to validate these potential applications. The aim of this work was to evaluate the effects of graphene on bone metabolism and the advantages of its use in implantology. A systematic review of literature was performed on PubMed, Web of Science and Scopus databases, and the articles investigating the role of graphene to functionalize dental implant surfaces and his interactions with the host tissue were analyzed.

**Keywords:** dental implant; graphene; graphene oxide; osteointegration; osteoblastic differentiation; titanium surface



## 1. Introduction

In recent years, the debate on the implant surface that can be most successfully used has increasingly intensified, replacing the earlier focus of researchers on implant shape [1–3]. One of the main reasons for the dispute is the comparison between the incidence rates of peri-implantitis in implants with a roughneck surface and those with a machined surface [4,5]. In this climate of fervent debate, many studies have been carried out to investigate the properties and outcomes of the various implant surfaces [6–8].

All implants are constituted of three basic components: the body of the implant (fixture), the abutment, and the abutment screw. A schematization of the implant fixture with the coating is presented in Figure 1. Macroscopically, implants can have various geometries, including threaded or tapered forms [7,9].

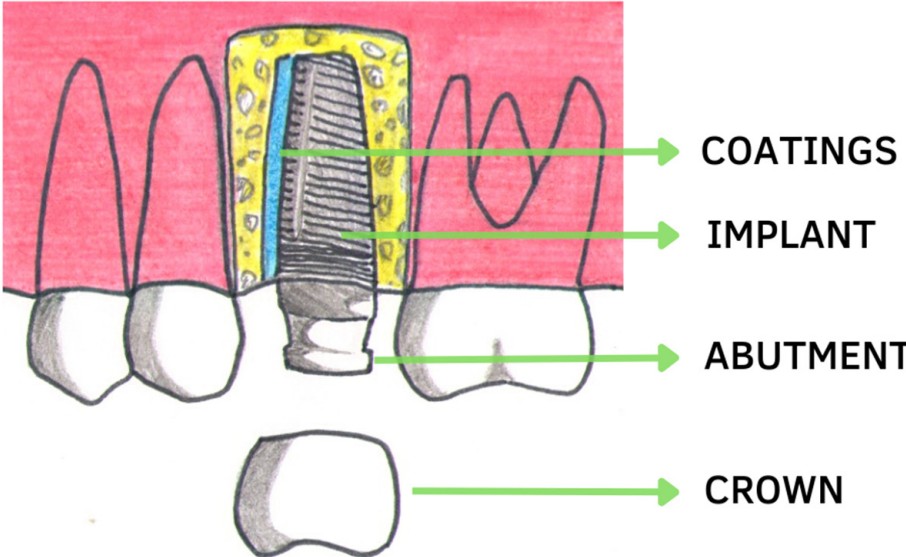

**Figure 1.** Components of dental implants and a representation of an implant fixture functionalized with coatings.

Osteointegration was defined 60 years ago by Branemark as a direct connection, structural and functional, between new bone and the surface of implant [10]. Current trials on dental implants, however, have as their objective the identification of the best system for enhancing the process of osseointegration, thereby allowing early loading [6]. Two factors are involved in the process of osseointegration: primary and secondary stability [11]. When the implant is inserted into the bone, certain areas of the surface come into direct contact with the bone. This contact determines the primary or mechanical stability, and depends on the shape of the implant, the quality of the bone and the preparation of the implant site [12]. Primary stability gradually decreases during the bone remodeling process and is completely replaced by biological stability (Figure 2). Secondary or biological stability is determined by the amount of new bone development at the bone-to-implant contact [13]. The studies by Osborn and Newesley showed that new bone formation occurs through two phenomena, distant and contact osteogenesis. In the first case, the deposition by osteoblasts and the subsequent mineralization takes place in a direction that goes from the periphery towards the implant, i.e., the bone gradually surrounds the screw [14]. In the second process, osseointegration occurs in the opposite direction, from the implant to the periphery. The apposition of new bone requires a continuous recall of cells from the bone and bloodstream towards the implant, since osteoblasts, after differentiation, are only capable of producing bone by apposition [14]. Once they are polarized, they produce ECM proteins, especially collagen, with the aim of giving a precise structure to the bone–implant interface, which, after calcification, turns into osteoid matrix and finally into bone tissue [14]. It should be noted that osteointegration is also linked to the concepts of osteoinduction and osteoconduction [13]. The former is related to the stimulation of osteoprogenitor cells to osteoblastic differentiation, a phenomenon that initiates osteogenesis, therefore "inducing" it. Osteoconduction, on the other hand, concerns the growth of the bone on a surface, therefore implying the existence of more or less osteoconductive surfaces, i.e., able to favor better or worse the adhesion and adaptation of the cells to the implant site. It can be seen that the direct anchorage (osseointegration) between the implant and the new bone, if maintained successfully and without the interposition of fibrous tissue (conversely, osteofibrointegration involves inflammatory reactions, bone resorption and implant failure), is nothing more than the concrete result of a previous osteoinduction and osteoconduction [13]. The osseointegration process and its quantity depend on the type of implant surface, which can have a geometry that attracts osteoblast cells; therefore, great

effort from researchers has been directed towards improving the capabilities of titanium implant surfaces [7,15].

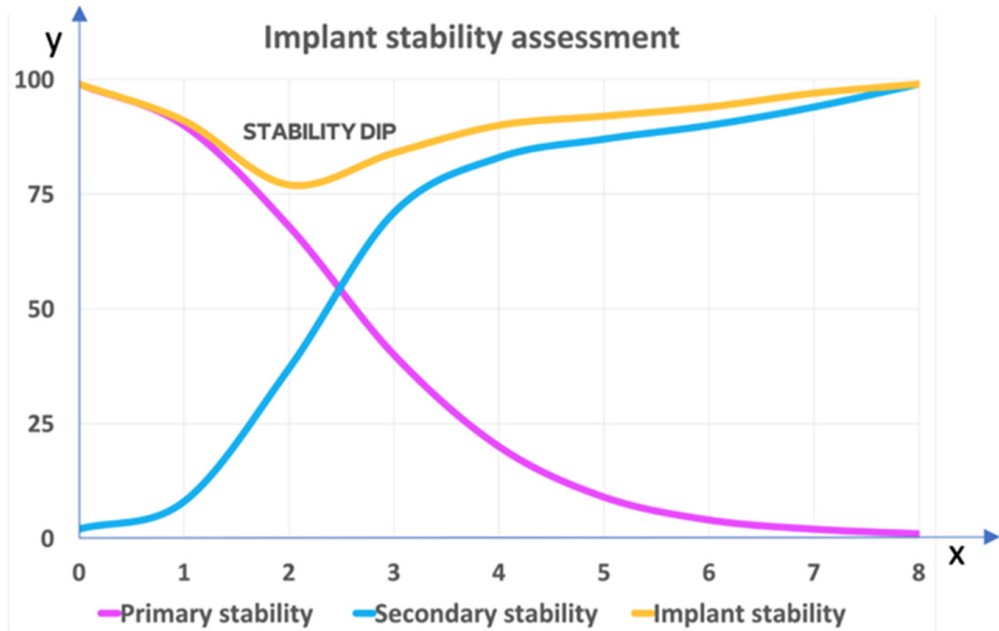

**Figure 2.** Implant stability assessment. X axis: weeks. Y axis: stability (%). Primary stability (purple) and secondary stability (blue) respectively decrease and increase, according to the curve in the graph. The yellow curve indicates total stability, with a dip after the first two weeks. The bone-to-implant contact (BIC) is a well-established technique to assess the degree of osseointegration and the speed of healing of dental implants.

Certain surface modifications may also include bioactive materials. The contact between the implant and the bone can be influenced by the shape of the implant, such as grooves, ridges, and tool marks [16]. The overall surface area that is available for osseointegration may be increased by the implant shape. Rougher surfaces can encourage bone cells to adhere, differentiate, and proliferate, resulting in an increase in bone formation and mineralization [17]. It has been demonstrated that rougher surfaces with an open structure promote quicker and more efficient osseointegration. Unfortunately, this rougher substrate's surface has a propensity to harboring microorganisms [16–18].

The majority of approaches aim to modify surface roughness. Machine work, blasting, laser etching, 3D printing, acid/alkaline etching, anodizing, and coatings can all cause surface roughness. Implant surfaces that have been machined typically have a roughness of less than 1 m. The roughest surfaces are those that have undergone plasma spraying and blasting. The roughness is determined by the sizes of the blasting particles. Polyelectrolytes do not change the surface roughness of acid-etched or machined titanium surfaces after sandblasting. There is disagreement on the ideal implant surface roughness that will have the optimum benefits on bone [19].

The surface composition has an Impact on hydrophilicity, which adds to the wettability and surface energy of the implant surface. Current surface modification approaches can increase hydrophilicity and surface area. Surface chemistry and wettability can both be changed by electrochemical functionalization [20].

In addition to modifying the surface using physical or chemical methods, it is possible to change its properties by adding other materials to the titanium surface, e.g., bioactive glasses, ceramics, etc. [21].

Different surface treatments have been developed to improve osseointegration, which allow the micro-topography to be roughened, such as machining, sandblasting, acid-etching, anodization, grit-blasting, and various coatings [22]. The surface treatment has the

purpose of increasing the contact area of the implant with the biological tissues, improving the osseointegration between the bone tissue and the implant. Already with a single thread, the degree of resistance to tensile and compressive force is greater than that of machined implants; the presence of micro-retention on the surface of the fixture allows for increase the tensile strength and torsion of the implant [22].

Some authors have demonstrated that macrophages, epithelial cells and osteoblasts have a high trophism towards rough surfaces [22–24]. An implant's surface roughness has the capacity to precisely select one population of cells and change their functions [22,25].

Sandblasting the titanium surface improves the biomechanical characteristics of the implant and helps to increase primary stability, enhancing the mechanisms of osseointegration [26].

Acid-etched surfaces are another subtractive technique that can be performed with different acids (sulfuric, hydrochloric) [25]. Following the good results provided by the two subtractive sandblasting and etching techniques, it was decided to combine their advantages in a single treatment, in order to obtain an SLA (sandblasted, large grit, acid-etched) surface: the first sandblasting phase determines a roughness that guarantees a strong mechanical fixation, the acid attack instead perfects the topographical conformation and helps to promote the adhesion of proteins, which is considered essential in the initial stages of bone healing [25,27].

Another method of surface treatment consists of additive techniques, coating the implant via titanium plasma spraying (TPS). This technology is currently used to increase surface roughness [28].

A lot of scientific work has been focused on bioactive surface coatings: these novel methods aim to imitate the metabolic environment and nanostructural organization of human bone (biomimetic effect) [7,29]. Experiments have been performed using a number of substances, medications, growth factors and proteins with the aim of producing innovative coatings [30]. One example of a process that is performed on the surface of dental implants is veneering with hydroxyapatite, which promotes the complete integration of the titanium with human bone tissues [31]. Additionally, bone morphogenetic proteins (BMPs), platelet-derived growth factor (PDGF), insulin-like growth factors (IGF-1 and 2), and a group of transforming growth factors (TGF-β family) could be used as bone-stimulating substances applied to the surface of titanium dental implants [32–35].

Short sequences of amino acids make up the biomolecules known as peptides [35]. Designing new implant surfaces has made use of certain peptides that promote cell adhesion in osseointegration or that have antibacterial properties like RGD peptide or human beta-defensins (HBDs) [29,36].

Implant surfaces can potentially include drugs that regulate the process of bone remodeling. In clinical conditions lacking bone support, such as resorbed alveolar ridges, incorporation of bone antiresorptive medications, such as biphosphonates, may be extremely significant [37]. Recent researches have demonstrated that adding biphosphonate to titanium implants enhances local bone density in the peri-implant area [38–40].

Graphene is a single layer of sp2-hybridized carbon atoms with a honeycomb lattice, and is a key component of fullerenes and carbon nanotubes. It has been attracting considerable interest in the physical, chemical and biomedical fields in recent years due to its mechanical properties, optimal electrical conductivity and very high surface area [41–43]. In particular, this material has been the subject of studies on osteogenesis, neurogenesis and biogenesis, showing strong biocompatibility and good stem cell differentiation in studies [44–46]. In addition, studies have shown graphene oxide (GO) to have antibacterial properties. GO is an oxidized version of graphene with numerous oxygen bonds on both accessible sides, such as carboxyl (-COOH), carbonyl (-C=O), and hydroxyl (OH). The presence of these groups enhances interactions with biomolecules and causes bacterial death without involving the cell. GO's antibacterial activity is linked to a variety of processes, including membrane stress, oxidative stress, entrapment, basal plane, and photothermal impact. The rough edges of GO nanolayers can physically disrupt bacterial membranes, leading to bacterial inactivation owing to internal matrix leaking [47–50].

*Porphyromonas gingivalis*, *Fusobacterium nucleatum*, *Tannerella forsythia*, and *Treponema denticola* seem to compose the majority of the peri-implant biofilm [51]. He et al. showed that graphene oxide decreased *F. nucleatum* and *P. gingivalis* proliferation [52]. Furthermore, Ghorbanzadeh et al. reported the use of graphene oxide (GO)-coated composites in drastically lowering the metabolic activity of *Streptococcus mutans*, a prominent cariogenic agent [53]. Another GO-CS-HA composite demonstrated lower S. aureus adhesion while also offering strong corrosion resistance and no cell toxicity [54]. GO has also been tested against *Candida albicans* in combination with curcumin and polyethylene glycol [55,56]. Graphene has also been coupled with curcumin for utilization as an antimicrobial photodynamic therapy photosensitizer agent for peri-implantitis treatment. The results showed that there was a great effectiveness in reducing biofilm development [57].

These properties of graphene can be exploited in implant surface design to overcome the problems associated with typical titanium-based dental implants. The aim of this paper is to evaluate the current and future applications of graphene with titanium in implantology by examining its possible advantages.

## 2. Materials and Methods

### 2.1. Protocol and Registration

The Preferred Reporting Items for Systematic Reviews and Meta-Analyses (PRISMA) guidelines were used in this systematic review [58].

### 2.2. Inclusion Criteria

All appropriate trials were assessed by two reviewers using the following inclusion criteria: (1) studies with human subjects; (2) studies in vitro; (3) open-access studies that other researchers can access for free; (4) articles in English language.

### 2.3. Exclusion Criteria

The exclusion criteria used in the search strategy were as follows: (1) animal studies; (2) systematic reviews; (3) meta-analyses; (4) narrative reviews; (5) letters; (6) books; (7) articles on implants not related to dentistry.

### 2.4. Search Processing

We searched PubMed, Scopus and Web of Science with a constraint on English-language papers from 8 February 2013 through 8 February 2023 that matched our topic. The following Boolean keywords were utilized in the search strategy: "Graphene Coating" AND "Dental Implant". These terms were chosen because they best describe the objective of our investigation, which was to evaluate the role of graphene as a material in dentistry useful in the prevention of bacterial infections and in supporting osseointegration.

## 3. Results

A total of 178 publications were identified from the PubMed (41), Scopus (57) and Web of Science (80) databases, resulting in 145 articles after the removal of duplicates (33). Analysis of the title and abstract resulted in the exclusion of 109 articles as being off topic, because they were not related to the use of graphene in implant dentistry. The writers successfully retrieved the remaining 36 papers and evaluated their eligibility. The approach resulted in the exclusion of 12 articles for being off topic. The evaluation finally resulted in the inclusion of 14 papers for qualitative analysis (Figure 3; Table 1).

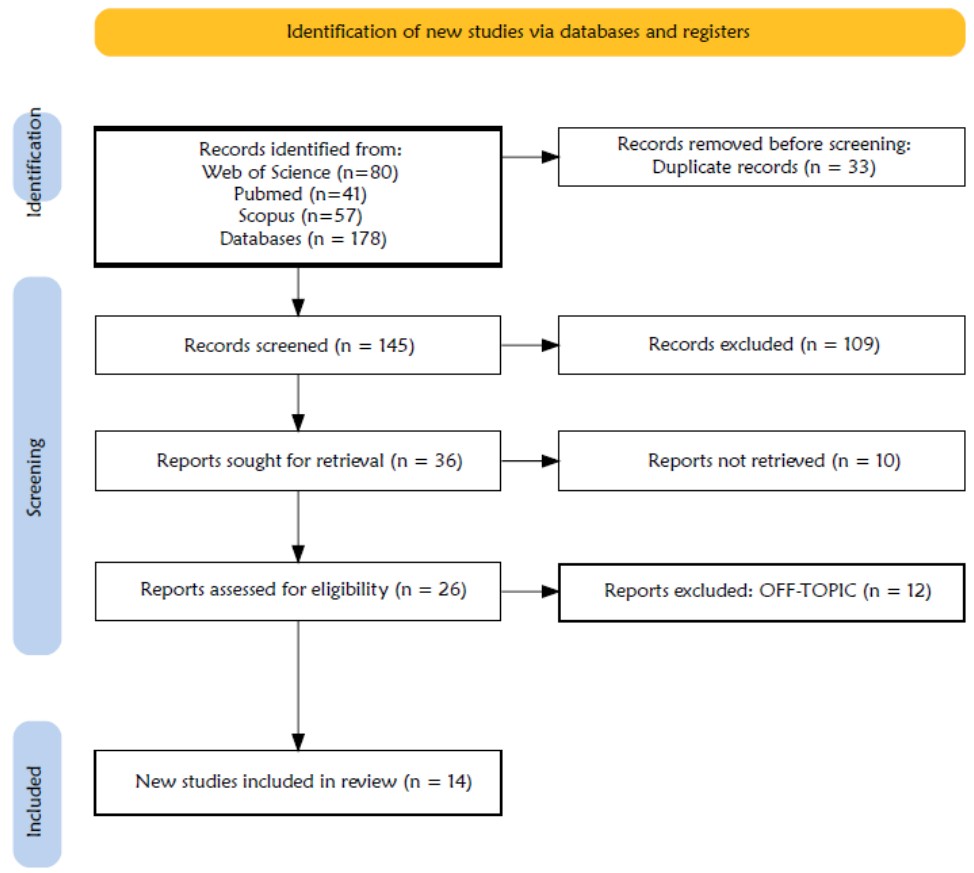

**Figure 3.** PRISMA flowchart diagram of the inclusion process.

**Table 1.** Studies included for qualitative analysis.

| Authors | Type of Study | Materials and Methods | Conclusions |
|---|---|---|---|
| Kulshrestha, S. et al., 2014 [59] | In vitro study | *S. mutans* MTCC 497 on plates of Graphene Nanocomposite/Zinc Oxide (GZNC) | - GZNC has antibacterial action against *S. mutans*<br>- GZNC treatment inhibits biofilm production<br>- GZNC as an implant coating reduces cytotoxicity |
| Ren, N. et al., 2017 [60] | In vitro study | Graphene oxide (GO) sheets produced by the modified Hummer's technique were combined with bioactive titanate on titanium implants (GO-Ti) before reduction (rGO-Ti). Cell proliferation of rat bone mesenchymal stem cells (rBMSCs) on them was assessed by mRNA expression and alkaline phosphatase activity. | Results revealed that Dexamethasone loaded surface (DEXdex-GO-Ti) performed superbly in increasing cell proliferation. In RMBSCs on DEX-GO-Ti, osteogenic differentiation-related proteins, mRNA, and calcium were all highly expressed. |
| Dubey, N. et al., 2018 [61] | In vitro study | - *S. mutans* and *Enterococus faecalis* on titanium plates coated with graphene<br>- Culture of human osteoblastic cells in contact with graphene-coated titanium plates | - Graphene-coated titanium:<br>1. Is cytocompatible<br>2. Induced the maturation of human osteoblasts<br>3. Increased mineralized matrix deposition compared with titanium alone<br>- GO was found to reduce the growth of Streptococcus Mutans and Enterococus Faecalis |

**Table 1.** *Cont.*

| Authors | Type of Study | Materials and Methods | Conclusions |
|---|---|---|---|
| Suo, L. et al., 2018 [62] | In vitro and in vivo study | Groups created for in vitro and in vivo evaluation: A. HA-Ti: Titanium + Hydroxyapatite (HA). B. GO/HA-Ti: group A coated with GO. C. CS/HA-Ti: group A coated with chitosan (CS) D. GO/CS/HA-Ti: group D coated by GO | Graphene oxide/chitosan/hydroxyapatite (GO/CS/HA)-coated titanium increases BMSC cell adhesion, proliferation, and differentiation in vitro. In addition, it demonstrated superior osseointegration during in vivo animal tests (rat tibia) |
| Rho, K. et al., 2019 [63] | In vitro study | Graphene-coated titanium with non-thermal atmospheric pressure plasma treatment | - Argon plasma treatment improves biocompatibility of titanium - Graphene oxide deposition by nonthermal plasma at atmospheric pressure enhances cell differentiation into osteoblasts, ensuring bone growth around the implant |
| Agarwalla, S.V. et al., 2021 [64] | In vitro study | Growth of C. albicans on graphene-coated grade 4 titanium plates was monitored for seven days. Uncoated titanium was the Control. | Graphene coating on titanium surface inhibits C. albicans biofilm formation due to its hydrophobic properties |
| Li, Q. and Wang, Z. 2020 [65] | In vitro and in vivo study | Evaluation of the behavior of rBMSC on acid-etched titanium SLA surfaces (control group) and on graphene-coated acid-etched titanium SLA surfaces. In addition, osteogenesis was evaluated in vivo (in rat femur). | The coating of GO: - Made the SLA surface more hydrophilic and capable of protein adsorption - Promoted adhesion, cell proliferation and osteogenic differentiation of BMSCs (activation of FAK/P38 pathway)High bone regeneration capacity was observed around GO-modified implants placed in rat femurs |
| Agarwalla, S.V. et al., 2019 [66] | In vitro study | *Streptococcus mutans, Enterococcus faecalis, Pseudomonas aeruginosa*, and *Candida albicans* biofilm development was assessed after 24 h on graphene coating titanium surfaces | For all species, titanium surfaces transferred two times with graphene (TiGD) offered superior quality while reducing the development of biofilm. The production of biofilms was shown to be reduced in correlation with enhanced hydrophobicity of graphene sheets. |
| Kang, M.S. et al., 2021 [67] | In vitro study | Atomic force microscopy (AFM), water contact angle, and Raman spectroscopy were used to analyze the physicochemical properties of rGO-coated Ti substrates. hMSCs were also cultivated on rGO-Ti, and their cellular characteristics, such as growth and osteogenic differentiation, were assessed. | By applying rGO evenly to Ti substrates the surface roughness and contact angle of Ti substrates could be reduced. After 7 days of incubation, rGO-Ti substrates greatly enhanced cell proliferation. |
| Lorusso, F. et al., 2021 [68] | In vivo study | Graphene-doped poly methyl methacrylate (PMMA) was compared to PMMA to determine water sorption, water solubility, and tolerance in rabbits using pyrogen test. | The levels of water sorption and solubility were very low in all of the testing samples. After the treatment, unaged graphene-doped PMMA specimens shown a stability in their physical and optical characteristics. Animal tests on the graphene-doped PMMA failed to produce pyrogens, an intradermal and systemic irritant. |

**Table 1.** *Cont.*

| Authors | Type of Study | Materials and Methods | Conclusions |
|---|---|---|---|
| Cao, X. et al., 2022 [69] | In vitro study | Anodic oxidation was used to prepare electrodeposition-loaded $TiO_2$ and GO nanotubes. Pure titanium disks was used as the control group and GO-coated titanium surface was used as the experimental group. | GO can modulate the cellular behavior of HGF on titanium surfaces. It also activates the MAPK signaling pathway to regulate HGF adhesion, spreading and migration, possibly by promoting TGF-β1 expression to promote HGF proliferation. |
| Shin, Y.C. et al., 2022 [70] | In vitro and in vivo study | Acid-etched SLA Ti (ST) implants were modified with rhBMP-2 and rGO. In vitro cell behaviors, in vivo osseointegration activity were evaluated among different groups, including ST (control), rhBMP-2-immobilized ST (BI-ST), rhBMP-2-treated ST (BT-ST) and rGO-coated ST (R-ST). | The titanium surface coated with rGO<br>- Has high biocompatibility and superior ability to absorb exogenous proteins<br>- Promotes cell growth and osteogenic differentiation without any osteogenic factors<br>- Accelerates osseointegration and dental tissue regeneration in vivo |
| Kwak, J.M. et al., 2022 [71] | In vitro and in vivo study | In vitro, BMSCs and Human Gingival Fibroblasts (HGFs) were seeded onto titanium discs, the surfaces of which had been treated in four different ways (SLA and/or GO).<br>In vivo, a rabbit tibia model is used to observe the effects of the four surface treatments on the osseointegration of titanium implants. | GO coating of implant surfaces promote cell adhesion, proliferation, osteogenic differentiation and osseointegration.<br>- Expression of ALP, RUNX2 and COL1A1 in titanium disc cells increased after ALS treatment and GO coating<br>- Cell proliferation on GO-coated titanium discs was 25% higher than on non-GO-coated titanium discs<br>- In the rabbit tibia study, it was seen that the GO-coated titanium implant had the highest BIC |
| Baheti, W. et al., 2023 [72] | In vitro study | Modified Ti implant surfaces were coated with GO, HA, HA-2%GO, and HA-5%GO by electrophoresis deposition and compared with uncoated Ti. Biological characteristics and osteogenic efficacy of in vitro-cultured rBMSCs. | The Ti surface's roughness and hydrophilicity were enhanced by the HA-GO nanocomposite coating. Cell adhesion and diffusion were improved on HA-GO-modified Ti surfaces compared to untreated Ti or Ti modified by HA or GO alone. Moreover, on surfaces treated with HA-GO, the proliferation and osteogenic differentiation of BMSCs in vitro were enhanced. |

## 4. Discussion

Graphene oxide represents a promising nanomaterial because of its exceptional physical and chemical qualities. Considerable research in recent years has concentrated on using graphene in biomedical applications such as tissue engineering, antimicrobial materials, and implants [73]. In this review, we investigated the literature on PubMed, Web of Science and Scopus, regarding the role of graphene to functionalize dental implant surfaces and its interactions with the host tissue.

Graphene is a single atomic sheet of sp2 hybridized carbon atoms arranged in a honeycomb lattice. Is the thinnest and strongest substance presented in nature [74]. It was originally effectively isolated in 2004 by Geim and Novoselov. Due to its exceptional qualities, including mechanical strength, elasticity, and electrical characteristics, graphene has attracted a lot of attention in research [75]. The two primary graphene derivatives are graphene oxide (GO) and reduced graphene oxide (rGO) [76]. Because of its great biocompatibility and low levels of toxicity, its hydro-solubility and reactive oxygen functional groups, studies have shown that graphene and GO may be used as supports in the biomedical sector for tissue regeneration, cell differentiation and proliferation also for enhancing the bioactivity and mechanical performance of biomaterials as well as serving

as a carrier for drug and biomolecules [68]. Graphene is a single atomic sheet of sp2 hybridized carbon atoms arranged in a honeycomb lattice. Is the thinnest and strongest substance presented in nature [74]. It was originally effectively isolated in 2004 by Geim and Novoselov. Due to its exceptional qualities, including mechanical strength, elasticity, and electrical characteristics, graphene has attracted a lot of attention in research [75]. The two primary graphene derivatives are graphene oxide (GO) and reduced graphene oxide (rGO) [76]. Because of its great biocompatibility and low levels of toxicity, its idro-solubility and reactive oxygen functional groups, studies have shown that graphene and GO may be used as supports in the biomedical sector for tissue regeneration, cell differentiation and proliferation also for enhancing the bioactivity and mechanical performance of biomaterials as well as serving as a carrier for drug and biomolecules [68].

According to a recent review by Liu et al., graphene has a potential role in oral disease treatment [73]: regarding restorative materials, it may be used for caries filling, since it may enhance the physicochemical and mechanical qualities of dental polymers and exhibit superior biocompatibility. It can be mixed with glass ionomer cements to enhance the composites' mechanical characteristics without having a negative impact on their aesthetic properties or their ability to release fluoride [77]. As shown by other authors, it can also enhance the quality of primer adhesion [78]. According to a recent review by Liu et al. Graphene has a potential role in oral disease treatment [73]: regarding restorative materials, it may be used for caries filling, since it may enhance the physicochemical and mechanical qualities of dental polymers and exhibit superior biocompatibility. It can be mixed with glass ionomer cements to enhance the composites' mechanical characteristics without having a negative impact on their aesthetic properties or their ability to release fluoride [77]. As shown by other authors, it can also enhance the quality of primers' adhesion [78].

Evidence suggests that graphene and GO exert an anticaries effect, preventing the development of *S. mutans* and *P. gingivalis*. In addition, graphene and GO can encourage human dental pulp stem cell (hDPSC) and periodontal ligament stem cell (PDLSC) differentiation and proliferation, which is helpful for the regeneration of dental pulp and periodontal ligament [73]. Graphene and its derivatives have shown considerable potential for the development of drug delivery systems, particularly for the administration of medications for targeted cancer therapy [73].

Unfortunately, there are currently just a few lines of support for the use of GO for this scope.

Graphene improves the antibacterial capabilities of ZnO nanoparticles, with an antibacterial effect that is significantly stronger than that of ZnO nanoparticles alone. The biofilm growth on the teeth was decreased by 85% with the GZNC coating [59]. According to this investigation, GZNC may work well against *S. mutans* as an antibacterial and antibiofilm agent. The nanocomposite also works well as a veneering agent for dental implants due to its low toxicity [59].

Bacterial adhesion must be prevented, since it is difficult to eliminate biofilm in the oral cavity once it has formed. By altering the implant surface, biofilm development can be prevented [79].

The minimal inhibitory concentration (MIC) of graphene nanocomposite/zinc oxide (GZNC) against *S. mutans* was found to be 125 $\mu$g mL$^{-1}$ in the study of Kulshrestha et al., whereas the minimum bactericidal concentration (MBC) was found to be 250 $\mu$g mL$^{-1}$ [59]. Graphene was found to diminish the growth of Streptococcus Mutans and Enterococcus Faecalis in the study of Dubey et al. [61], and to decrease the growth of Candida albicans in the study of Agarwalla et al. [64]. However, the antibacterial activity in the graphene oxide sample in the investigation ok Rho et al. was quite weak [63]. Moreover, titanium's biocompatibility was improved by argon plasma treatment, and non-thermal plasma deposition of GO at atmospheric pressure encouraged bone development around the implant by promoting osteoblast cell differentiation [63]. According to a study of Shin et al., increasing the concentration of rGO on titanium surfaces results in rougher surfaces that are more able to absorb exogenous proteins, which in turn promotes cell proliferation and osteogenic

differentiation [70]. Furthermore, in the study of Kwak et al., the surface roughness of SLA discs was higher than that of titanium discs [71]. After GO coating, the surface roughness decreased slightly, but no significant difference was observed. Furthermore, the expression of alkaline phosphatase (ALP, a marker for osteoblastic differentiation) in cells on titanium discs increased after SLA treatment and GO coating [71]. Cell proliferation on GO-coated titanium discs was 25% higher than that on non-GO coated titanium discs [71]. Notably, the expression of ALP, Runt-related transcription factor 2 (RUNX2) and collagen type I-$\alpha$1 (COL1A1) is upregulated in cells on GO-SLA-Ti discs [71]. These authors expanded their rabbit tibia study and found that GO-coated titanium implant had the highest bone-to-implant contact ratio (BIC), followed by GO-SLA-Ti, SLA-Ti and Ti [71]. For the GO-Ti and SLA-Ti groups, there were statistically significant variations in BIC ratio. The BIC ratios between the four implant groups were comparable at 4 weeks, though [71]. In the study by Li et al., bone marrow mesenchymal stem cells (BMSCs) grown on the surface of GO-modified material produced higher ALP than those grown on the surface of SLA material. This demonstrates the ability on the part of GO to induce early osteogenic differentiation of BMSCs [65]. The FAK/P38 pathway is activated by GO modification-induced BMSC osteogenic differentiation. Moreover, SLA/GO implants demonstrated outstanding in vivo osseointegration performance, because TGF and BMP2 were expressed at the gene level, the RUNX2 factor was activated, and PGE2 production was increased [65].

Suo et al. observed that titanium coated by graphene oxide/chitosan/hydroxyapatite (GO/CS/HA-Ti) had greater bioactivity in terms of enhanced adhesion, proliferation, and differentiation of BMSCs during in vitro cytological evaluation, and it demonstrated superior osseointegration during in vivo animal testing [62]. On titanium surfaces, GO can alter how human gingival fibroblasts (HGFs) behave cellularly [69]. According to research, GO regulates HGF adhesion, diffusion, migration, and proliferation through activating the MAKP signaling pathway, presumably by encouraging TGF-1 expression [69]. Furthermore, it was mentioned in the study by Kang et al. that rGO-Ti substrates supported hMSC osteogenic differentiation as well as proliferation [67]. These results are primarily explained by the unique properties of rGO, such as its hydrophilic nature and electrical conductivity, thus improving cell adhesion, protein absorption from serum, and cell–cell or cell–matrix signaling [67]. It has been demonstrated in the literature that titanium surfaces coated with hydroxyapatite (HA) and GO increase hydrophilicity and cause mouse bone marrow mesenchymal stem cells to differentiate into osteoblasts (BMSCs) [72]. Even in the research of Agarwalla et al., samples in which the graphene has been transferred dry without the use of risky agents exhibit an increase in titanium's hydrophilicity [66]. Titanium surfaces with GO coatings are more hydrophilic than SLA surfaces, on average [65]. Moreover, the GO-modified surface interacts with proteins through electrostatic interactions and encourages protein adsorption [65].

In the research by Ren et al., GO was applied as a coating to titanium foils as a drug delivery system to promote osteo-differentiation and cell proliferation of rat bone mesenchymal stem cells (rBMSC) [60]. Dexamethasone loaded on GO coupled with bioactive titanate on Ti implants (DEX-GO-Ti) and Dexamethasone loaded on rGO coupled with bioactive titanate on Ti implants (DEX-rGO-Ti) both allowed for improved adsorption and long-lasting release of dexamethasone (DEX) [60]. For DEX-GO-Ti substrates as opposed to DEX-Control and DEX-rGO-Ti substrates, a significantly greater proliferation rate of rBMSCs was attained. Moreover, rBMSCs differentiated more osteogenically on DEX-GO-Ti and DEX-rGO-Ti substrates than on DEX-Control substrates [60]. As a result, the titanium surface's DEX-loaded GO coating controlled the bioactivity of Ti implants, paving the way for new developments in dentistry [60].

Biomaterials derived from polymethylmethacrylate (PMMA), and in particular PMMA doped with graphene (GD-PMMA), were tested by Lorusso et al., who evaluated the osseointegration capacity of GD-PMMA [68]. In their study, 18 PMMA and 18 GD-PMMA implants were placed in the femoral knee joint of male rabbits [68]. All implants integrated well into the bone, but the GD-PMMA titanium surfaces were shown to improve osseoin-

tegration in rabbit femurs. Furthermore, the authors suggested that further in vitro and in vivo animal studies are needed to evaluate a potential clinical use for dental implant applications [68,80].

*Streptococcus mutans* growth has been demonstrated to be inhibited by GZNC [66], while studies by Dubey et al. [61] and Agarwalla et al. [66] indicated that GO also reduced the growth of Streptococcus Mutans and Enterococcus Faecalis and Candida albicans. Rho et al., in contrast, claimed that the antibacterial activity of GO was only moderate [63]. In addition, non-thermal plasma deposition of GO at atmospheric pressure would encourage bone growth around the implant by boosting osteoblastic cell differentiation, which would increase the biocompatibility of titanium [63].

Increased rGO concentration on titanium surfaces produces rougher surfaces that may absorb exogenous proteins, which in turn encourages cell proliferation and osteogenic differentiation, according to research by Shin et al. [70] and Kwak et al. [71]. ALP, RUNX2, and COL1A1 expression also seems to be enhanced by rGO [71]. Several studies have demonstrated the ability of GO to induce early osteogenic differentiation of BMSCs by activating the FAK/P38 pathway and increased adhesion capacity, proliferation of BM-SCs [62,65]. In addition, GO can regulate adhesion, migration and proliferation of HGF [69] and hMSCs [66,67,72]. In the research of Ren et al., GO was applied as a titanium sheet coating as a drug delivery system to promote osteodifferentiation and cell proliferation of rBMSCs [60]. Improved DEX adsorption and prolonged DEX release were made possible by DEX-GO-Ti and DEX-rGO-Ti, respectively [60]. A considerably higher rate of rBMSC proliferation was attained for DEX-GO-Ti substrates as compared to DEX-Control and DEX-rGO-Ti substrates. Moreover, on DEX-GO-Ti and DEX-rGO-Ti substrates compared to DEX-Control substrates, rBMSCs differentiated more osteogenically [60]. As a result, the titanium surface's DEX-filled GO covering controlled the bioactivity of Ti implants, opening the door for future advancements in dentistry [60].

Lorusso et al. [68] evaluated implants made of PMMA, specifically GD-PMMA. The study's objective was to assess GD-osseointegration PMMA's capacity before considering it as a possible material for dental implant devices [68]. In their research, male rabbits with femoral knee joints had 18 PMMA and 18 GD-PMMA implants inserted [68]. All implants fused successfully with the bone, although research on rabbit femurs revealed that GD-PMMA titanium surfaces enhanced osseointegration. More in vitro and in vivo animal investigations, according to the authors, are required in order to assess the potential clinical utility of dental implant applications [68,80].

## 5. Conclusions

Graphene Oxide represents a promising nanomaterial because of its exceptional physical and chemical qualities. From our results graphene coatings may considerably increase osteogenic differentiation of bone marrow mesenchymal stem cells in vitro by the regulation of FAK/P38 signaling pathway and can encourage in vivo the osteointegration of dental implants. However, these potential applications need further studies to be validated, especially in humans. The addition of surface roughness from GO coatings to implant surfaces was found to be stable, non-reactive, and conducive to cell adhesion, diffusion, and proliferation. For addressing several significant issues, the use of GO in implant veneers appears promising. First off, germs found on the tissues surrounding the implant and graphene oxide's antibacterial properties are two of the primary causes of implant failure. Furthermore, several studies have shown that GO can help with osseointegration. Second, GO has the ability to bind biomolecules and active ingredients that may aid further improve osseointegration and quicken the healing process.

Based on the body of research, we draw the conclusion that GO coatings hold significant promise for preserving a healthy balance between a coated dental implant's capacity to prevent biofilm formation and its capacity to incite a positive cellular response.

**Author Contributions:** Conceptualization, A.M.I., A.D.I., G.M., F.P., A.P. and C.D.P.; methodology, G.P., F.P. and A.M.; software, F.I., G.P. and G.D.; validation, F.P., F.I. and G.D.; formal analysis, A.D.I., F.P., A.P., F.I. and G.D.; resources, A.D.I., A.M.I., G.M., C.D.P. and A.M.; data curation, G.M., G.D., F.I., G.P. and C.D.P.; writing—original draft preparation, A.D.I., A.M.I., F.P., F.I. and G.D.; writing—review and editing, A.P., G.M., G.P. and A.M.; visualization, G.P., C.D.P., A.M. and A.P.; supervision, A.D.I., A.M.I., F.P., G.M., A.P., F.I. and G.D.; project administration, A.P., F.I. and G.D. All authors have read and agreed to the published version of the manuscript.

**Funding:** This research received no external funding.

**Institutional Review Board Statement:** Not applicable.

**Informed Consent Statement:** Not applicable.

**Data Availability Statement:** Not applicable.

**Conflicts of Interest:** The authors declare no conflict of interest.

## Abbrevations

| | |
|---|---|
| ALP | Alkaline Phosphatase |
| BIC | Bone-to-Implant Contact |
| BMPs | bone morphogenetic proteins |
| BMSCs | Bone Marrow Mesenchymal Stem Cells |
| COL1A1 | Collagen type 1-$\alpha$1 |
| DEX | Dexamethasone |
| DEX-GO-Ti | Dexamethasone loaded on GO coupled with bioactive titanate on Ti implants |
| DEX-rGO-Ti | Dexamethasone loaded on rGO coupled with bioactive titanate on Ti implants |
| GD-PMMA | Graphene-doped polymethylmethacrylate |
| GO | Graphene Oxide |
| GO/CS/HA-Ti | Titanium coated by graphene oxide/chitosan/hydroxyapatite |
| GZNC | Graphene Nanocomposite/Zinc Oxide |
| HA | Hydroxyapatite |
| HBD | Human beta defensins |
| hDPSC | human dental pulp stem cell |
| HGFs | Human Gingival Fibroblasts |
| hMSCs | human Mesenchymal Stem Cells |
| IGF | insulin-like growth factors |
| MBC | Minimum Bactericidal Concentration |
| MIC | Minimal Inhibitory Concentration |
| PDGF | platelet-derived growth factor |
| PDLSC | Periodontal ligament stem cells |
| PMMA | Polymethylmethacrylate |
| rBMSC | rat Bone Mesenchymal Stem Cells |
| rGO | reduced graphene oxide |
| RUNX2 | Runt-related transcription factor 2 |
| SLA | Sandblasted, Large grit, Acid-etched |
| TPS | Titanium plasma-sprayed |

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
