# Peer review of "Potential of Graphene-Functionalized Titanium Surfaces for Dental Implantology: Systematic Review"

_coatings, doi:10.3390/coatings13040725_

Round 1
Reviewer 1 Report
In the present work, the authors described the main implant surface modification methods and their effects on bone formation, analyzed the rusults of the recent studies on graphene coating, and draw a conclusion that graphene coating has a promising future in dental implant based on its biological effects. The topic is closely related to clinical problem, the reference analysis is appropriate. However, the clarity and contextual relevance of this review need to be improved, and the selected reference need to be updated. The following issues should be addressed before being accepted for publication.
1. In this manuscript, a large amount of space is spent to describe the methods of implant surface roughness and hydrophilicity and their effects on bone formation. However, it has little relevance to the topic of graphene-functionalized titanium surface, and the context description needs to be further optimized.
2. In lines 329 to 336 of the manuscript, the author should use the third person rather than the first person to describe.
3. In lines 127, bacterial infections were mentioned for the first time, but it is not described in the previous background.
4. Some of the references selected in this manuscript have been published for a long time, which is not conducive to the readers' understanding of the latest development status. They need to be updated.
Author Response
See at the attached file

Reviewer 2 Report
Dear authors!
The question of the implant surface in relation to the process of osseointegration, as well as postoperative complications, is very interesting and relevant. The chosen study design allows to assess the state of the issue on the basis of the literature used, as well as draw conclusions about the prospects for its development, which corresponds to the purpose. However, some comments and questions appeared in the course of reviewing the research material, which, from my point of view, are worth paying attention to.
1. The statement about the increased interest of researchers in the most successful implant surface given in the text fragment [32-33] should be supported by scientific works in which the analysis of literary data was carried out to determine the degree of the issue being studied (literary review, systematic analysis, meta-analysis, etc.). In this case, these scientific works cannot confirm or refute the mentioned judgment, since this was not the purpose of these studies. I recommend excluding these literary references.
2. In the text fragment [34-36] the literary reference №4 should also be either deleted or replaced with a more appropriate one, since it is a case report and does not describe the state of the issue of the influence of the implant surface type on the frequency of complications (peri-implantitis).
3. The arguments in the text fragment [74-83] should be supported by appropriate literary references.
4. The sentence "The review protocol was registered at PROSPERO under the unique number" on lines [117-118] is almost identical to the sentence on line [120]. One of them should be excluded.
5. It is worth noting that the main advantages of systematic reviews are transparency and reproducibility, which allows to minimize subjectivity and bias. Unfortunately, I could not find information about the progress of your research in PROSPERO under the specified registration number (402053). Tell me, please, maybe you mistakenly entered the wrong number?
6. From my point of view, the "Materials and methods" section is not fully presented, since the details given in it do not allow us to reproduce this study exactly. I recommend complementing this section in accordance with the items presented in the PRISMA 2020 checklist https://systematicreviewsjournal.biomedcentral.com/articles/10.1186/s13643-021-01626-4/tables/2. For example, in addition to information about inclusion criteria, it is worth adding information about exclusion criteria.
7. In the "Materials and methods" section, in addition to the content part, it is worth paying attention to the approach of information presentation. So, when conducting a systematic review, the eligibility criteria are determined before search processing, so I recommend swapping the items "Search Processing" and "Inclusion criteria".
8. The "Results" section is also not fully presented. It is not entirely clear which articles were analyzed, there is no information about the characteristics of the included studies, the results of individual studies and etc. I recommend complementing this section in accordance with the items presented in the PRISMA 2020 checklist https://systematicreviewsjournal.biomedcentral.com/articles/10.1186/s13643-021-01626-4/tables/2.
9. Please, explain why paragraphs 3.1 and 3.2 were added to the "Discussion" section. From my point of view, the provided information is not suitable for the subject and purpose of the study. Also, this information cannot be compared with search keywords presented in the article. Unfortunately, it is impossible to verify this information, since there is no information about the included studies in the "Results" section.
10. The designations on Figure 3 must correspond to the description provided in the text fragment [147-149]. Thus, it is worth making changes either to the text of the publication, or to supplement the designations to the image in accordance with the information provided in the text.
11. I would advise to shorten the "Conclusions" section so that the presented text corresponded clearly to the stated purpose of the study.
12. A systematic review implies the selection and study of all available articles, that is why it is considered to be comprehensive. As I mentioned earlier, this approach allows to give the most objective assessment of the issue under study. After reviewing the list of references, I would like to note that the manuscript has 18 self-citations out of 71 references and that means more than 20% of references belong to the same authors. The issue of self-citation is very sensitive and has an ethical nature. Please explain the reason for the excessive quoting. Do these literary references relate to the articles included in the study, if not, what kind of links are they? Unfortunately, I could not verify this, since there is no information about the included studies in the "Results" section.
13. It is also worth paying attention to the design of the article and other errors:
- New paragraphs should start with a red line;
- [line 118] there is a missing space "number.The»;
- [line 170] the word "research" must begin with a capital letter.
Thus, based on the arguments given above, at this stage I have to recommend rejecting the submitted article, as it requires serious revision.
Author Response
See at the attached file

Reviewer 3 Report
An interesting topic that needs to be addressed in the near future. however, as a reviewer, I have a few comments
Introduction
Line 84
Expansion of the implant surface. you can also mention 3 D printing. An implant is printed from two materials, then one is dissolved and then we get a very extensive surface,
Figure 1- no units on X and Y axes are X months and Y strength or percentage of stability? please explain it. ifsi % then how is it meirzone.
lines 103
there is no connection between the previous part and this paragraph.
I would add 1-2sentences. In addition to modifying the surface using physical or chemical methods, it is possible to change its properties by adding other materials to the titanium surface, e.g. bioactive glasses, ceramics, etc. for example:
Raszewski, Zbigniew & Chojnacka, Katarzyna & Mikulewicz, Marcin. (2022). Preparation and characterization of acrylic resins with bioactive glasses. Scientific Reports. 12. 16624. 10.1038/s41598-022-20840-1.
Other applications include the use of graphene.
Material and Methods
Line 122
We searched PubMed, Scopus and Web of Science with a constraint on English-language papers from 8 February 2013 through 8 February 2023 that matched our topic. Better to use no personal form Pubmed, Scoupus and Web have been reviewed for English-language articles…….
Line 137
Analysis of the title and abstract resulted in the exclusion of 109 articles- it would be good to add why so many articles were excluded Did they describe something else, were they not reviewed????
All in all, I would add the Results part, and finally the discussion?
Discussion
Line 187
3.2 Surfaces Coatings: future perspective- add the space
A bit strange. In Pud Med, have you been looking for implant surfaces, graphene and in your conclusions you describe half-pages about traditional preparation of implant surfaces and amino acids?
Line 207
3.3 GRAPHENE: an overview of its applications- add the space, You used lowercase letters in the previous paragraphs, so it applies here as well
Line 220
idro-solubility- what dose it mean/ Hydro- solubility ( mistake from Italian) or explain this term, please.
Line 223
In the last period of time, more and more research show that nanoparticles of materials that are safe in macro composition (for example, ZnO, SiO2), show a completely different effect on the human body. Therefore, it may be similar in the case of graphene.
I would add here a few articles that determine whether graphene is biologically inert, and not just one publication [55] saying that the material placed inside PMMA is safe. Well, PMMA has a contact with tissues, and it has been known for 50 years that it is an intertwin material. It's a suggestion.
Line 272
Dubey et al, but ref [60] has different authors?
Line 294
Even in the study of Li et al., BMSCs grown on the surface of the GO-modified material produced higher ALP than those grown on the surface of the SLA material, demonstrating the clear promotion of early osteogenic differentiation of BMSCs by GO-BMSCs what dose it mean? Explain this aberration.- The second thing is that you repeat the same abbreviation twice in one sentence, can you write it differently? Using full name Bone Marrow Mesenchymal Stem Cells…
Line 300
GO/CS/HA-Ti- please explain this aberration, DEX-GO-Ti and DEX-rGO-Ti?
Line 329
To date, plants derived from polymethylmethacrylate (PMMA) and in particular PMMA doped with graphene (GD-PMMA) are being tested- plants or parts?
Line 330
The aim of the study is to evaluate the osseointegration capacity of GD-PMMA and therefore to use it as a potential material for dental implant devices????- The purpose of your research was to determine the graphene on the surface of the implant or PMMA you need to fix it a bit??? This is already plagiarism because the whole sentence is taken from another article...? you must change it
References
Please correct the literature as there are no correct citations, There is a cite icon in Pub Med, there to get the correct citation format if you are not sure how to quote this article
Line 405
INCHINGOLO, F.; PARACCHINI, L.; DE ANGELIS, F.; CIELO, A.; OREFICI, A.; SPITALERI, D.; SANTACROCE, L.; GHENO, E.; PALERMO- please use the small letters
Line 414
Albrektsson T, Sennerby L. State of the art in oral implants. J Clin Periodontol. 1991 Jul;18(6):474-81. doi: 10.1111/j.1600-051x.1991.tb02319.x. PMID: 1890231
Line 426
Goggles Scholar [CITACE] Dynamic aspects of the implant-bone interface
JF Osborn - in Materials and systems in dental implants, 1980 - cir.nii.ac.jp
Uložit Citovat Počet citací tohoto článku: 278 Související články Všechny verze (počet: 3)- this is not a true citation it should be
Osborn JF, Newesely H. Dynamic aspects of the implant bone interface. In: Heimke G, ed. Dental implants: materials and systems. München. Carl Hanser Verlag 1980:111-23.- but this article is not existing on the web side….
Line 463
M. Poot H. S. J. van der Zant Nanomechanical Properties of Few-Layer Graphene Membranes:
Appl. Phys. Lett. 92, 063111 (2008); https://doi.org/10.1063/1.2857472
Line 471
Park SY, Park J, Sim SH, Sung MG, Kim KS, Hong BH, Hong S. Enhanced differentiation of human neural stem cells into neurons on graphene. Adv Mater. 2011 Sep 22;23(36):H263-7. doi: 10.1002/adma.201101503. Epub 2011 Aug 8. PMID: 21823178.
Line 487
Sandblasted and Acid Etched Titanium Dental Implant Surfaces Systematic Review and Confocal Microscopy Evaluation – PubMed.
Cervino G, Fiorillo L, Iannello G, Santonocito D, Risitano G, Cicciù M. Sandblasted and Acid Etched Titanium Dental Implant Surfaces Systematic Review and Confocal Microscopy Evaluation. Materials (Basel). 2019 May 30;12(11):1763. doi: 10.3390/ma12111763. PMID: 31151256; PMCID: PMC6600780
Line 573
Dubey N, Ellepola K, Decroix FED, Morin JLP, Castro Neto AH, Seneviratne CJ, Rosa V. Graphene onto medical grade titanium: an atom-thick multimodal coating that promotes osteoblast maturation and inhibits biofilm formation from distinct species. Nanotoxicology. 2018 May;12(4):274-289. doi: 10.1080/17435390.2018.1434911. Epub 2018 Feb 6. PMID: 29409364.
Line 600
Baheti W, Lv S, Mila, Ma L, Amantai D, Sun H, He H. Graphene/hydroxyapatite coating deposit on titanium alloys for implant application. J Appl Biomater Funct Mater. 2023 Jan-Dec;21:22808000221148104. doi: 10.1177/22808000221148104. PMID: 36633270.
Good luck in further research
Author Response
See at the attached file

Round 2
Reviewer 2 Report
Dear authors!
Thank you for the work done which clarified some aspects and improved the quality of your manuscript. However, the following questions remain:
1. It is worth noting that in the literary references quoting lines [33-34] the question of graphene using possibility in dentistry is raised. Unfortunately, in this context, the references are irrelevant because they do not consider the state of the issue of the increased relevance of implant surface in comparison with implant shape. From my point of view, they should be replaced with more suitable ones or it is worth rephrasing the proposal and backing it up with appropriate references.
2. In revised version of your manuscript under “Figure 1” there is two almost identical graphs, one of them should be excluded.
3. In your article a submission number was identified, however your article was rejected by PROSPERO, so it was not registered in this system. The given registration data does not allow to judge about the progress of your research. That is incorrect so I recommend removing this information.
4. You have indicated that the evaluation includes a final 14 papers for qualitative analysis. However, in the “Results” section are given references to 16 publications which does not correspond to the information above. Thus, the question of the included studies remains open. It may be worth presenting the information about the included studies more clearly, for example, in the form of a table, so that the reader can easily identify the selected articles.
5. The text fragment [278-282] is of a reference nature and does not affect the relevance and state of the issue, so I recommend deleting it. It is also worth noting that paragraphs 1 and 2 in the "Discussion" section describe the state of the issue rather than interpret the received results, so it would be more logical to transfer this information to the "Introduction" section.
